# ASGen: Answer-containing Sentence Generation to Pre-Train Question Generator for Data Augmentation in Question Answering

## Abstract

Numerous machine reading comprehension (MRC) datasets often involve manual annotation, requiring enormous human effort, and hence the size of MRC data remains significantly smaller than that of those available for unsupervised learning, limiting the generalization capability. To overcome this issue, a new approach, which can generate synthetic question-and-answer data from large corpora such as Wikipedia, has been recently proposed. Such synthetic data can be utilized as additional data to pre-train the main MRC model before fine-tuning it using real, existing MRC data. However, the quality of generated questions and answers is still far from being satisfactory, so previous work introduced a pre-training technique for the question generator by pre-training on the generation of the next sentence that follows a paragraph. However, the next sentence that follows a paragraph may have little relevance to the questions or answers from within the paragraph, and thus it is not the ideal candidate for pre-training question generation. In response, we propose a novel method called Answer-containing Sentence Generation (ASGen). Our approach is composed of multiple stages, involving two advanced techniques, (1) dynamically determining $K$ answers from a given document and (2) pre-training the question generator using the task of generating the answer-containing sentence. We evaluate the question generation capability of our method by comparing the BLEU score with existing methods and test our method by fine-tuning the MRC model using the downstream MRC data after training on synthetic data. Experimental results show that our approach achieves outperforms existing methods achieving new state-of-the-art results on SQuAD question generation and increases the performance of the state-of-the-art MRC models across a range of datasets such as SQuAD-v1.1, SQuAD-v2.0, KorQuAD, and QUASAR-T with no architectural modifications to the original MRC model.

## 1 Introduction

Machine reading comprehension (MRC), which finds an answer to a given question from given paragraphs called context, is an essential task in natural language processing. With the use of high-quality human-annotated datasets for this task, such as SQuAD-v1.1 (Rajpurkar et al., 2016), SQuAD-v2.0 (Rajpurkar et al., 2018), and KorQuAD (Lim et al., 2019), researchers have proposed MRC models, often surpassing human performance on these datasets. These datasets commonly involve finding a short snippet within a paragraph as an answer to a given question.

However, these datasets require a significant amount of human annotation to create pairs of a question and its relevant answer from a given context. Often the size of the annotated data is relatively small compared to that of data used in other unsupervised tasks such as language modeling. Hence, researchers often rely on the two-phase training method of transfer learning, i.e., pre-training the model using large corpora from another domain in the first phase, followed by fine-tuning it using the main MRC dataset in the second phase.

Most state-of-the-art models for MRC tasks involve such pre-training methods. Peters et al. (2018) present a bidirectional contextual word representation method called ELMo, which is pre-trained on a large corpus, and its learned contextual embedding layer has been widely adapted to many

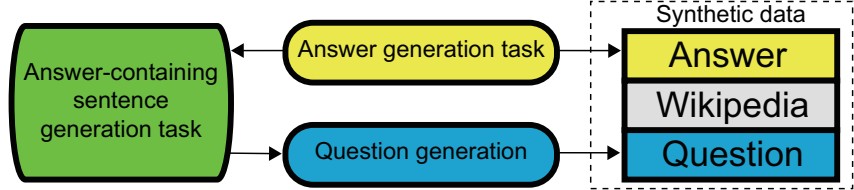

Figure 1: Overview of our Answer-containing Sentence Generation (ASGen) method.

other MRC models. Devlin et al. (2019a) show that pre-training with a masked language model on a large corpus and then fine-tuning on a downstream dataset results in significant performance improvements.

However, pre-training on another domain task and then fine-tuning on a downstream task may suffer from performance degradation, depending on which pre-training task is used in the first phase. For example, Yang et al. (2019) show that the pre-training task of next sentence classification decreases performance on the downstream MRC tasks. To handle this problem, generating synthetic data similar to the those of a downstream task is crucial to obtain a properly pre-trained model. Recently, researchers have studied a model for generating synthetic MRC data from large corpora such as Wikipedia. This is essentially a form of transfer learning, by training a generation model and using this model to create synthetic data for training the MRC model, before fine-tuning on the downstream MRC dataset.

Golub et al. (2017) suggest a two-stage synthesis network that decomposes the process of generating question-answer pairs into two steps, generating a fixed number ($K$) of answers conditioned on the paragraph, and question generation conditioned on the paragraph and the generated answer. Devlin et al. (2019b) introduced a pre-training technique for the question generator of this method by pre-training on the generation of next-sentence that follows the paragraph.

However, choosing a fixed number ($K$) of candidate answers from each paragraph will lead to missing candidates if $K$ is too small, and will lead to having lower-quality candidates if $K$ is too big. Moreover, the next sentence generation task is not conditioned on the answer, despite the answer being a strong conditional restriction for question generation task. Also, the next sentence that follows a paragraph may have little relevance to the questions or answers from within the paragraph, and hence is not the ideal candidate for pre-training question generation.

To address these issues, we propose Answer-containing Sentence Generation (ASGen), a novel method for a synthetic data generator with two novel processes, (1) dynamically predicting $K$ answers to generate diverse questions and (2) pre-training the question generator on answer-containing sentence generation task. We evaluate the question generation capability of our method by comparing the BLEU score with existing methods and test our method by fine-tuning the MRC model on downstream MRC datasets after training on the generated data. Experimental results show that our approach outperforms existing generation methods, increasing the performance of the state-of-the-art MRC models across a wide range of MRC datasets such as SQuAD-v1.1, SQuAD-v2.0, KorQuAD, and QUASAR-T (Dhingra et al., 2017) without any architectural modifications to the MRC model.

## 2 PROPOSED METHOD

This section discusses the details of our proposed ASGen method. ASGen consists of a BERT-based generative model (BertGen) and answer-containing sentence generation pre-training (AS). First, we will describe how BertGen model generates synthetic data from Wikipedia. Next, we will explain the novel components of our methods and how we pre-trained the question generator in BertGen based on them. BertGen encodes paragraphs in Wikipedia with two separate generation networks, the answer generator and the question generator.

**Answer Generator.** As shown in Fig. 2-(1), we generate the number of answer candidates $K$ for a given context without the question by applying a fully connected feed-forward layer on the contextual embedding of classification token "[CLS]". To make the contextual embeddings and to predict

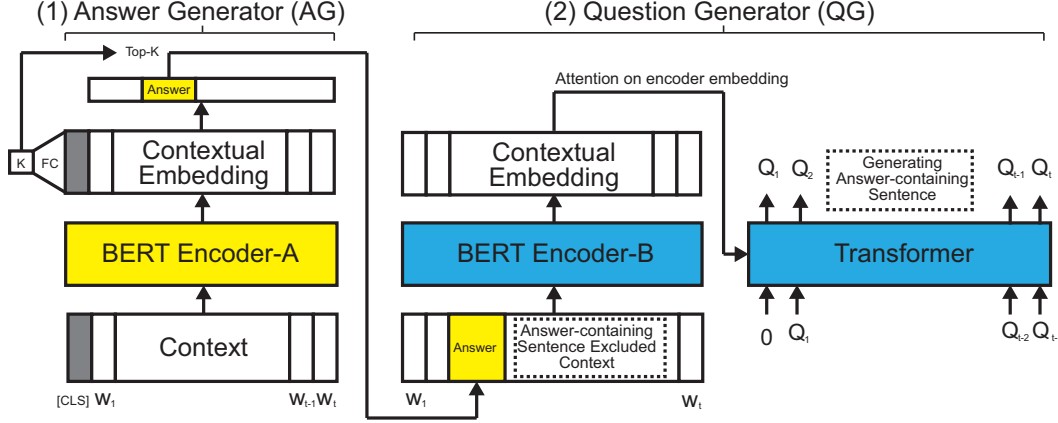

Figure 2: Architecture of ASGen (BertGen+AS), (1) Answer Generator and (2) Question Generator. In the case of answer-containing sentence generation task (AS), the question generator takes the answer and the context without the answer-containing sentence as input and generates the answer-containing sentence.

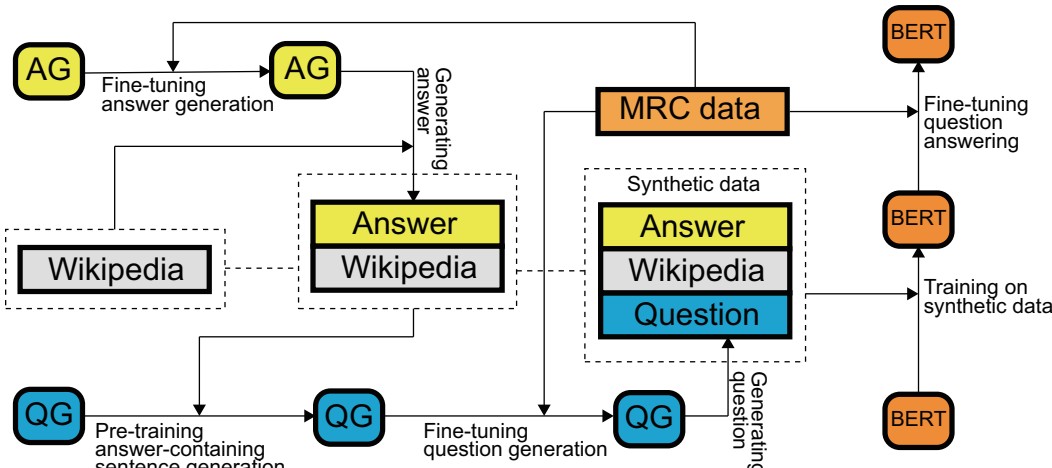

Figure 3: Detailed procedure of generating the synthetic MRC data from Wikipedia using ASGen.

answer spans, we utilize a BERT (Devlin et al., 2019a) encoder (Fig. 2-BERT Encoder-A). Depending on the predicted number $K$, we select the $K$ top candidate answer spans from the context. As shown in Fig. 2-(2), we use the $K$ selected candidate answer spans as input to the question generator.

**Question Generator.** Next, as shown in Fig. 2-(2), we generate a question conditioned on each answer predicted from the answer generator. Specifically, we pass as input to a BERT encoder the context and an indicator for the answer span location in the context (Fig. 2-BERT Encoder-Q). Next, a Transformer decoder (Vaswani et al., 2017) generates the question word-by-word based on the encoded representation of the context and the answer span. For pre-training such a question generator on an answer-containing sentence generation task, we exclude the answer-containing sentence from the original context and train the model to generate the excluded sentence given the modified context and the answer span as input.

Finally, we generate questions and answers from a large corpus, e.g., all the paragraphs in Wikipedia in this paper. After generating such data, we train the MRC model on the generated data in the first phase and then fine-tune on the downstream MRC dataset (such as SQuAD) in the second phase. In this paper, we use BERT as the default MRC model, since it exhibits state-of-the-art performance in many MRC datasets.

## 2.1 DYNAMIC ANSWER PREDICTION

The most natural method for humans to create a question-answer pair from a given context is to select the answer first and then create a corresponding question. In this situation, we conjecture that a human is more likely to choose as an answer a phrase that is "answer-like", such as keyphrases, nouns, dates, names, etc. There may be several answers in the context that are likely to be selected by humans as answers, especially if the context is lengthy or if it contains multiple nouns, dates, names, etc.

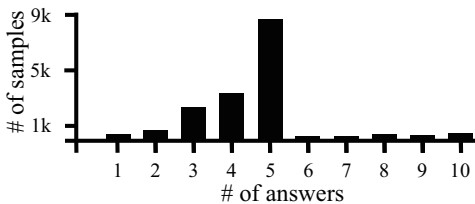

Figure 4: Distribution of the number of answers in each paragraph in SQuAD-v1.1.

For example, the context "Barack Hussein Obama II is an American attorney and politician who served as the 44th president of the United States from 2009 to 2017" can have as possible answers "Barack Hussein Obama", "44th", "United States", "2009 to 2017", etc. As shown in Fig. 4, to see these characteristics, we examine the distribution of the number of answers in the SQuAD dataset and hypothesize that there exists an underlying pattern in the number of answers that occur in a context. The conventional method to generate multiple answers from a context is to draw a fixed number ($K$) of answers. However, this approach can generate low-quality answers if $K$ is too big, and it can impact the number and diversity of the generated answers if $K$ is too small.

Therefore, we predict the number of answers $K$ in a given context $W = \{\mathbf{w}_t\}_0^T$ using regression as,

$$\{\mathbf{w}_t^{enc}\}_{t=0}^T = \text{BERT Encoder-A}(\mathbf{W})_t,$$
$$K = \lfloor f_k(\mathbf{w}_0^{enc}) \rfloor,$$

where $T$ is the number of word tokens in the context with position 0 reserved for classification token '$[CLS]$', and $f_k$ represents a fully connected unit with two hidden layers that have hidden dimensions equal to $H$ and 1, respectively, where $H$ is the hidden dimension of BERT Encoder-A.

To calculate the score $s_i$ for start index $i$ of a predicted answer span, we compute the dot product of the encoder output with a trainable start vector $\mathbf{S}$. For each start index $i$, we calculate the span end index score $e_{i,j}$ for end index $j$ in a similar manner with a trainable end vector $\mathbf{E}$, but conditioned on $i$, i.e.,

$$\{s_i\}_{i=0}^T = \mathbf{S} \circ \mathbf{w}_i^{enc}$$
$$\{e_{i,j}\}_{i,j=0}^{T,T} = \mathbf{E} \circ f_s(\mathbf{w}_j^{enc} \oplus \mathbf{w}_i^{enc}),$$

where $f_s$ represents a fully connected layer with hidden dimension $H$ and $\oplus$ indicates the concatenation operation. For training, we use the mean squared error loss between $K$ and ground-truth number of answers. We also use cross-entropy loss on the $s_i, e_{i,j}$ and ground truth start/end of the answer span for each token. Predicting the number of answers and predicting the span are jointly trained to minimize the sum of their respective losses.

During inference, we choose the $K$ top answer spans with the highest score summation of start index score and end index score, i.e.,

$$A^{span} = \{(i,j) \mid 1 \leq i < T \text{ and } i \leq j < T\},$$
$$a_k = \max(\{a \mid \#\{(i,j) \mid (i,j) \in A^{span} \text{ and } s_i + e_{i,j} \geq a\} = K\}),$$
$$A_k^{span} = \{(i,j) \mid (i,j) \in A^{span} \text{ and } s_i + e_{i,j} \geq a_k\}.$$

The $K$ selected answer spans $A_k^{span}$ are then given to the question generator as input in the form of an indication of the answer span location.

## 2.2 PRE-TRAINING QUESTION GENERATOR

In order to generate questions conditioned on different answers that may arise in a context, we generate a question for each of the $K$ answers. Devlin et al. (2019b) previously proposed to pre-train this generation model with an unsupervised task that generates the next sentence following a given paragraph to improve generation performance. We identify several issues with this approach. The

final question generation task has the form of sentence generation given an answer and a context, while the next-sentence generation has no answer component. The next-sentence generation task is not conditioned on the answer, despite the answer being a strong conditional constraint for the question generation task. Also, the next sentence that follows a paragraph may have little relevance to the questions or answers from within the paragraph, and hence is not the ideal candidate for pre-training question generation.

To address these issues, we modify the context to exclude the sentence containing our previously generated answer and pre-train our generator on the task of generating this excluded answer-containing sentence, conditioned on the answer and the modified context.

Specifically, we exclude answer-containing sentence $S^{ans}$ while leaving the answer and modify the original context $D$ to $D^{ans}$ as

$$S^{start} = \{p \mid \text{sentence start index} = p\},$$

$$S^{ans} = \{(p, q, i, j) \mid \max(\{p|p{\leq}i\}), \min(\{q|q{\geq}j\}), (i, j) \in A_k^{span}, p \in S^{start}, q \in S^{start}\},$$

$$D^{ans} = [D_{[:p]}; D_{[i:j]}; D_{[q:]}], (p, q, i, j) \in S^{ans}.$$

Note that we change $S^{ans}$ to not exclude the answer-containing sentence in the case of fine-tuning on the question generation, i.e.,

$$S^{ans} = \{(p, q, i, j)|p = i, q = j\}.$$

Afterwards, we pass the previously generated answer to the sequence-to-sequence generation model as a segmentation encoding $M^{ans}$ that identifies the answer part within the context, i.e.,

$$M^{ans} = [\mathbf{m}_0 * p; \mathbf{m}_1 * (j - i); \mathbf{m}_0 * (T - q)], (p, q, i, j) \in S^{ans},$$

where $\mathbf{m}_0$ and $\mathbf{m}_1$ indicate trainable vectors corresponding to segmentation id 0 and 1, respectively. Here we tag the segmentation id for each word in the context as 0 and each word in the answer as 1. $A * B$ indicates the operation of concatenating vector $A$ for $B$ many times.

Next, we generate answer-containing sentence embedding $W^g = \{\mathbf{w}_t^g\}_0^T$ using a Transformer sequence-to-sequence model (the encoder part is initialized with BERT) as

$$\mathbf{w}_t^g = \text{Transformer Decoder}(\{\mathbf{w}_i^g\}_{i=0}^{t-1}, \text{BERT Encoder-Q}(D^{ans}, M^{ans})).$$

Finally, we calculate the loss of the generation model with cross-entropy over generated sentence words, i.e.,

$$\{\mathbf{w}_t^o\}_{t=0}^T = \{\text{Softmax}(\mathbf{w}_t^g E)\}_{t=0}^T,$$

$$\mathbb{L} = -\left(\sum_{t=1}^{T}\sum_{i=1}^{D} \mathbf{y}_{t,i}\log(\mathbf{w}_{t,i}^o) + (1 - \mathbf{y}_{t,i})\log((1 - \mathbf{w}_{t,i}^o))\right) / T,$$

where $\mathbf{y}$ indicates a ground-truth one-hot vector of the answer-containing sentence word (the question word in the case of fine-tuning), $D$ is the vocabulary size, and $E \in \mathbb{R}^{d \times D}$ represents a word embedding matrix shared between the BERT Encoder-Q and the Transformer decoder.

In this manner, we pre-train the question generation model using a task similar to the final task of conditionally generating the question from a given answer and a context.

## 3 EXPERIMENTAL SETUP

**Pre-training Dataset.** To build the dataset for answer-containing sentence generation tasks (AS) and the synthetic MRC data for pre-training the downstream MRC model, we collect all paragraphs from the entire English Wikipedia dump (Korean Wikipedia dump for KorQuAD) and synthetically generate questions and answers on these paragraphs. We apply extensive filtering and cleanup to only retain high quality collected paragraphs from Wikipedia. Detailed pre-processing steps for obtaining the final Wikipedia dataset can be found in the supplemental material.

Using the answer generator in ASGen (BertGen+AS), we generate 43M answer-paragraph pairs (Full-Wiki) from the final Wikipedia dataset for pre-training on answer-containing sentence generation. For ablation studies on pre-training approaches, we also sample 2.5M answer-paragraph

pairs (Small-Wiki) from Full-Wiki and 25K answer-paragraph pairs (Test-Wiki) to evaluate the pre-training method. Finally, using the question generator in ASGen (BertGen+AS), we generate one question for each answer-paragraph pair in Full-Wiki and create the final synthetic MRC data containing 43M triples of a paragraph, its question and its answer.

**Benchmark Datasets.** In most MRC datasets, a question and a context are represented as a sequence of words, and the answer span (indices of start and end words) is annotated from the context words based on the question. Among these datasets, we choose SQuAD as the primary benchmark dataset for question generation, since it is the most popular human-annotated MRC dataset. SQuAD-v1.1 (Rajpurkar et al., 2016) consists of crowd-sourced questions and answers based on contexts from Wikipedia articles. We compare our question generation capability with existing question generation methods such as UniLM (Dong et al., 2019). For fair comparison, we split the training set of SQuAD-v1.1 data into our own training and test sets, and keep the original development set as our dev set, as previously done in Du et al. (2017), Kim et al. (2019), and Dong et al. (2019). We call this dataset as Test Split1[1]. We also evaluate on the reversed dev-test split, called Test Split2.

To evaluate the effect of generated synthetic MRC data, we evaluate the fine-tuned MRC model on the downstream MRC dataset after training on the generated synthetic data. We perform this on SQuAD-v1.1 and SQuAD-v2.0 (Rajpurkar et al., 2018). We also evaluate on KorQuAD (Lim et al., 2019) which is another dataset created with the same procedure as SQuAD-v1.1 for Korean language. To show that our generated data is useful for other MRC datasets, we fine-tune and test the MRC model on QUASAR-T (Dhingra et al., 2017) which is large-scale MRC dataset, after training on the synthetic data that generated from SQuAD-v1.1.

**Implementation Details.** For the answer generator, we use BERT (Devlin et al., 2019a) and two fully connected layers to predict the number of answers $K$. For the BertGen model, we use pre-trained uncased BERT (Base) as encoder and 12 layers of Transformer as decoder. For the generation of unanswerable questions as in SQuAD-v2.0, we separate unanswerable cases and answerable cases and train separate generation models. For the final MRC model, we use BERT (Large) which is the state-of-the-art model on multiple datasets with all official hyper-parameters. We use the Mecab (Kudo, 2006) tokenizer for Korean to separate postposition words which do not exist in English.

**Comparison of the Pre-training Method.** We compare our question generation pre-training method, which is pre-training on answer-containing sentence generation task (AS), with a method from Devlin et al. (2019b), which is pre-training on next-sentence generation task (NS), and with a method from Golub et al. (2017), which only trains question generation on final MRC dataset. We reproduced these methods on BertGen as they were described in their original work for comparison. Note that 'BertGen+AS' is equivalent to 'ASGen'. We generate synthetic data from Wikipedia using these approaches which are trained on the target downstream MRC datasets except for QUASAR-T. In the case of QUASAR-T, we use synthetic data which is generated by ASGen trained on SQuAD-v1.1. To check the effectiveness of our method on downstream MRC tasks, we evaluate our generated data on SQuAD-v1.1, SQuAD-v2.0, KorQuAD and QUASAR-T by training state-of-the-art models (BERT and BERT+CLKT[2]) on generated data followed by fine-tuning on the train set for each dataset. The structure of 'BERT + CLKT' model is the same as that of original BERT except that the model is pre-trained for the Korean language. Due to the absence of common pre-trained BERT for Korean, we used this model as a baseline to demonstrate the effectiveness of our method.

## 4    QUANTITATIVE RESULTS

**Dynamic Answer Prediction.** We conducted an experiment to demonstrate the performance of our method in generating the number of answers in a given context. As shown in Table 1, in the case of fixed $K$, the mean absolute error from the ground-truth $K^{gt}$ is the smallest at $K^{pred} = 5$ and the values are 1.92 and 0.99 for Test Split1 and Test Split2, respectively. Thresholding on the sum of the start and end logits with a fixed threshold value which minimizes the mean absolute error results in

---

[1]We use the identical splitting of SQuAD data provided by UniLM from its publicly available website (`https://github.com/microsoft/unilm`)

[2]'BERT+CLKT with ASGen' model can be found as 'BERT-CLKT-MIDDLE' on the leaderboard (`https://korquad.github.io/KorQuad%201.0`)

Table 1: Mean absolute error of prediction $K^{pred}$ with respect to ground-truth $K^{gt}$. The results are obtained on SQuAD Test Split1 and Test Split2.

| Model | Mean Absolute Error | |
|---|---|---|
| | Test Split1 | Test Split2 |
| Thresholding on Answer Logits | 2.31 | 1.12 |
| Fixed-$K$ ($K^{pred} = 5$) | 1.92 | 0.99 |
| Dynamic-$K$ Answer Prediction | 1.24 | 0.76 |

Table 2: Comparison of BLEU-4 scores with existing models on SQuAD Test Split1 and Test Split2. ASGen (Large) has 24 layers of encoder and decoder. Those models with * are reproduced.

| Question Generation Model | BLEU-4 score | |
|---|---|---|
| | Test Split1 | Test Split2 |
| Du et al. (2017) | 12.3 | - |
| Zhao et al. (2018)* | 12.8 | 14.9 |
| ASs2s (Kim et al., 2019) | 16.2 | - |
| Zhao et al. (2018) | - | 16.4 |
| UniLM (Dong et al., 2019) | 22.1 | 23.8 |
| ASGen (Full-Wiki) | 21.5 | 24.7 |
| ASGen (Full-Wiki) (Large) | 25.4 | 28.0 |

an error of 2.31 and 1.12, respectively in the two splits. In contrast, our answer generator generates a more appropriate number of answers than the fixed $K$ approach, by reducing the mean absolute error between the ground-truth $K^{gt}$ and the prediction $K^{pred}$ of 1.24 and 0.76, respectively for the two splits.

**Question Generation.** To evaluate our question generator, we fine-tune the model on both Test Split1 and Test Split2, after pre-training answer-containing sentence generation on Full-Wiki. As shown in Table 2, ASGen outperforms existing methods by 0.9 BLEU-4 score on Split2, 24.7 for ASGen vs. 23.8 for UniLM. Moreover, our final question generation model, ASGen (Large), outperforms existing methods by a large margin in BLEU-4 score on both splits, 25.4 for ASGen (Large) vs. 22.1 for UniLM for Split1 and 28.0 for ASGen (Large) vs. 23.8 for UniLM for Split2.

To show the effectiveness of our answer-containing sentence pre-training task (AS), we compare between various pre-training tasks. As shown in Table 3, AS is shown to perform better than NS, e.g. 21.5 vs. 18.2 and 24.7 vs. 19.7 in the two splits, respectively. Note that conditioning on a given answer has only a small effect on AS, e.g. 19.4 vs 19.5. This implies the performance gain is largely due to pre-training on the answer-containing sentence generation task rather than conditioning on a given answer.

We also compare the BLEU-4 scores between before and after applying AS on other existing question generation models. We reproduce Zhao et al. (2018) and use the official code of Dong et al. (2019). As shown in Table 4, AS consistently improves the performance of other question generation models with no architecture changes or parameter tuning.

**Downstream Task Performance.** We conduct experiments by training MRC models on the synthetic data generated by ASGen from Wikipedia before fine-tuning the model on the downstream dataset to show the effectiveness of our synthetic data generation. For each dataset, the MRC model is pre-trained on the corresponding generated synthetic data and fine-tuned on the downstream data. As shown in Table 5, the MRC model pre-trained on the synthetic data generated by ASGen shows an improvement of 1.9 F1 score on SQuAD-v1.1, 4.0 F1 score on SQuAD-v2.0, and 0.5 F1 score on KorQuAD from the state-of-the-art baseline models. Moreover, using the synthetic data generated from ASGen shows better performance than using the synthetic data generated from 'BertGen+NS' on both SQuAD-v1.1 and SQuAD-v2.0 downstream data.

**Effects of MRC and Synthetic Data Size.** Fig. 5 shows the effects of synthetic data with respect to the size of the synthetic and real MRC data. In Fig. 5-(a), where we fix the size of synthetic data as

Table 3: Comparison among pre-training methods of the question generator in ASGen, i.e, without pre-training, pre-training on NS, pre-training on AS, pre-training on AS without conditioning on a given answer. Note that we use Small-Wiki for comparison of pre-training except those entries including "(Full-Wiki)".

| Model + pre-training method | BLEU-4 score | | |
| --- | --- | --- | --- |
| | Wikipedia Test-Wiki | SQuAD v1.1 Test Split1 | SQuAD v1.1 Test Split2 |
| BertGen | - | 14.4 | 16.9 |
| BertGen + NS | 1.4 | 17.5 | 19.2 |
| BertGen + AS wo/ condition to a given answer | 6.8 | 19.4 | 21.2 |
| BertGen + AS [ASGen] | 7.1 | 19.5 | 22.2 |
| BertGen + NS (Full-Wiki) | 3.4 | 18.2 | 19.7 |
| BertGen + AS (Full-Wiki) [ASGen] | 11.3 | 21.5 | 24.7 |

Table 4: Effects of pre-training on answer-containing sentence generation (AS) on other existing methods. We use Small-Wiki data to pre-train existing models. Those models with * are reproduced.

| Model + pre-training method | BLEU-4 score | | |
| --- | --- | --- | --- |
| | Wikipedia Test-Wiki | SQuAD v1.1 Test Split1 | SQuAD v1.1 Test Split2 |
| Zhao et al. (2018)* | - | 12.8 | 14.9 |
| Zhao et al. (2018)* + AS | 6.8 | 14.0 | 16.2 |
| UniLM (Dong et al., 2019) | - | 22.1 | 23.8 |
| UniLM (Dong et al., 2019) + AS | 8.9 | 22.9 | 24.5 |

43M, the F1 score of MRC model pre-trained on the synthetic data generated by ASGen consistently outperforms that of BertGen+NS. In particular, performance difference becomes apparent for a small size of real MRC data, while the performance gap diminishes for a large size. Such a gap may become insignificant for a sufficient size of real MRC data, but for the current size of SQuAD data (87K in total) AS still improves the performance.

As shown in Fig. 5-(b), we also conducted experiments by training the MRC model using a different amounts of generated synthetic data for the same number of iterations, while using the full size of real SQuAD data. The total number of training steps for all data sizes is kept the same as that of 10M synthetic data. A larger size of generated data consistently gives better performance.

**Transfer Learning to Other Datasets.** In this experiment, we first fine-tune ASGen using SQuAD-v1.1, and using synthetic data generated by this ASGen, we train BERT MRC model. Afterwards, we fine-tune BERT for the downstream MRC task using QUASAR-T, in order to verify that the data generated in this manner is useful for other MRC datasets. QUASAR-T has two separate datasets, one with short snippets as context, and the other with long paragraphs as context. As shown in Table 6, training with our synthetic data is shown to improve the F1 score by 2.2 and 1.7 for the two cases, respectively.

## 5 QUALITATIVE RESULTS

**Comparison of Question Generation.** We qualitatively compare the generated questions after pre-training with NS and AS to demonstrate the effectiveness of our method. For the correct answer "49.6%" as shown in the first sample in Table 7, NS omitted "Fresno", which is a critical word to make the question specific, while AS's question does not suffer from this issue. Note that the word "Fresno" occurs in the answer-containing sentence. This issue also occurs in the second sample, where NS uses the word "available" rather than the more relevant words from the answer-containing sentence, but AS uses many of these words such as "most" and "popular" to generate contextually rich questions. Also, the question from NS asks about "two" libraries, while the answer has "three" libraries, showing the lack of sufficient conditioning on the answer. The third sample also shows that

Table 5: Comparison of EM/F1 scores of fine-tuned MRC model on SQuAD v1.1, SQuAD v2.0, and KorQuAD dev sets using their corresponding synthetic data for pre-training.

| Model | Synthetic data | SQuAD v1.1 | | SQuAD v2.0 | | KorQuAD | |
|---|---|---|---|---|---|---|---|
| | | EM | F1 | EM | F1 | EM | F1 |
| BERT | - | 83.9 | 90.9 | 78.8 | 81.8 | - | - |
| | BertGen | 85.1 | 91.4 | 80.9 | 83.9 | - | - |
| | BertGen + NS | 85.6 | 92.3 | 81.5 | 85.1 | - | - |
| | BertGen + AS [ASGen] | 86.3 | 92.8 | 82.5 | 85.8 | - | - |
| BERT + CLKT | - | - | - | - | - | 87.1 | 94.5 |
| | BertGen + AS [ASGen] | - | - | - | - | 87.8 | 95.0 |

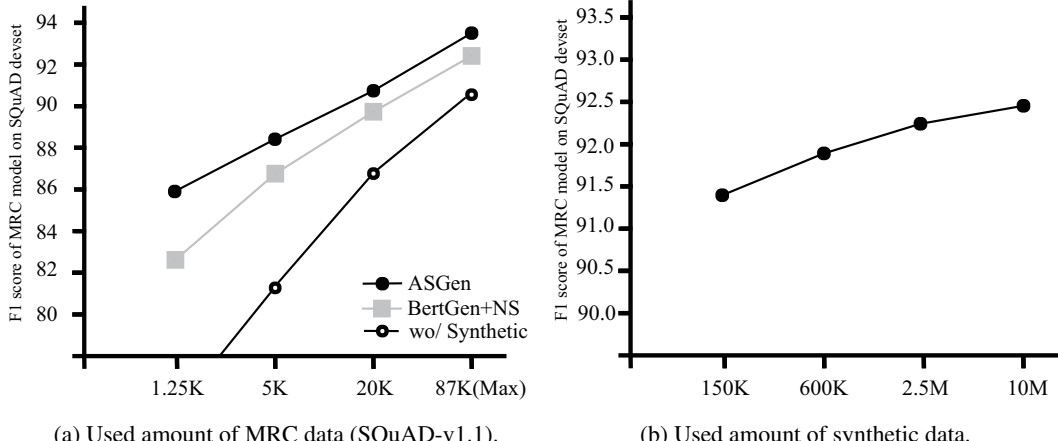

(a) Used amount of MRC data (SQuAD-v1.1).  (b) Used amount of synthetic data.

Figure 5: Comparison of F1 of the BERT model on SQuAD-v1.1 dev set with different data sizes.

AS draws more context-related questions than NS by including the exact subject "TARDIS" to use for the corresponding answer in a similar vein.

## 6 RELATED WORK

**Machine Reading Comprehension.** For MRC tasks, a large number of datasets have been proposed, most often focused on finding an answer span for a question from a given paragraph. Popular and fully human-annotated datasets include SQuAD-v1.1 (Rajpurkar et al., 2016), SQuAD-v2.0 (Rajpurkar et al., 2018), KorQuAD (Lim et al., 2019), and HotpotQA (Yang et al., 2018). However, these datasets are relatively small with around 100K samples each, which is far smaller than those datasets used for unsupervised tasks such as language modeling.

**Question Generation.** Question generation methods have been actively studied for various purposes including data augmentation in question answering. Du et al. (2017) proposed an attention-based model for question generation by encoding sentence-level as well as paragraph-level information. Song et al. (2018) introduced a query-based generative model to jointly solve question generation and answering tasks. Kim et al. (2019) separately encoded the answer and the rest of the paragraph for proper question generation. Zhao et al. (2018) utilized a gated self-attention encoder with a max-out unit to handle long paragraphs. Our proposed method (AS) can further improve the question generation quality of these methods by pre-training them with an answer-containing sentence generation task.

**Transfer Learning.** Pre-training methods have been increasingly popular in natural language processing to obtain contextualized word representations. Open-GPT (Radford et al., 2018), BERT (Devlin et al., 2019a), XLNet (Yang et al., 2019), and UniLM (Dong et al., 2019) use a Transformer module (Vaswani et al., 2017) to learn different styles of language models on a large dataset fol-

Table 6: Comparison of EM/F1 scores of the BERT fine-tuned on QUASAR-T dataset. The synthetic data are generated from ASGen trained on SQuAD-v1.1.

| Model | Synthetic data | Short(Dev) | | Short(Test) | | Long(Dev) | | Long(Test) | |
|-------|----------------|------|------|------|------|------|------|------|------|
| | | EM | F1 | EM | F1 | EM | F1 | EM | F1 |
| BERT | - | 74.3 | 78.6 | 74.1 | 77.8 | 72.1 | 75.6 | 72.1 | 74.8 |
| | ASGen (SQuAD-v1.1) | 76.5 | 80.1 | 76.5 | 80.0 | 74.2 | 77.4 | 73.8 | 76.5 |

Table 7: Examples from SQuAD-v1.1 dev set demonstrating generated questions. We compare our method (AS) with NS. Colored Text indicates a given answer.

| | |
|---|---|
| Context | The 2010 United States Census reported that Fresno had a population of 494,665. The population density was 4,404.5 people per square mile. (1,700.6km). The racial makeup of Fresno was 245,306 ( 49.6% ) White, 40,960 (8.3%) African American, 8525 (1.7%) Native American ... (omit) |
| Question by NS | What percent of the population is White? |
| Question by AS | What percentage of the Fresno population is White? |
| Context | (omit) ... According to the American Library Association, this makes it the largest academic library in the United States, and one of the largest in the world. Cabot Science Library, Lamont Library, and Widener Library are three of the most popular libraries for undergraduates to use, with ... (omit) |
| Question by NS | Which two libraries are available for undergraduates to use? |
| Question by AS | What are the three most popular libraries for undergraduates? |
| Context | (omit) ... He fled from Gallifrey in a stolen Mark I Type TARDIS "Time and Relative Dimension in Space" time machine which allows him to travel across time and space. The TARDIS has a "chameleon circuit"...(omit) |
| Question by NS | What does the doctor refer to? |
| Question by AS | What does the TARDIS stand for? |

lowed by fine-tuning on the downstream task. While our approach is similar to these approaches, our pre-training task for question generator generates answer-containing sentences to learn better representations for the question generation task.

**Synthetic Data Generation.** Subramanian et al. (2018) show that neural models generate better answers than using off-the-shelf tools for selecting named entities and noun phrases. Golub et al. (2017) proposed to separate the answer generation and the question generation. This model generates questions conditioned on generated answers, and then they evaluate the quality of the synthetic data by training an MRC model with them before fine-tuning on SQuAD. Inspired by the observations from previous studies, we improved the performance of answer generation and question generation by using a newly designed models as well as a novel pre-training technique.

## 7 CONCLUSIONS

We propose two advanced training methods for generating high-quality and diverse synthetic data for MRC. First, we dynamically choose the $K$ top answer spans from an answer generator and then generate the sentence containing the corresponding answer span as a pre-training task for the question generator. Using the proposed methods, we generate 43M synthetic training samples and train the MRC model before fine-tuning on the downstream MRC dataset. Our proposed method outperforms existing questions generation methods achieving new state-of-the-art results on SQuAD question generation and consistently shows the performance improvement for the state-of-the-art models on SQuAD-v1.1, SQuAD-v2.0, KorQuAD, and QUASAR-T datasets without any architectural modification to the MRC model.

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

## A    Details of Wikipedia Preprocessing

To build the answer-containing sentence generation data and the synthetic MRC data, we collect all paragraphs from all articles of the entire English Wikipedia dump (Korean Wikipedia dump for KorQuAD) and generate questions and answers on these paragraphs. We apply extensive filtering and cleanup to only retain the highest-quality paragraphs from Wikipedia.

To filter out low-quality obscure pages, we remove all pages that received less than 200 cumulative page-views including all re-directions in a 2-month period. In order to calculate the number of page-views, official Wikipedia page-view dumps were used. Of the 5.4M original Wikipedia articles, filtering by page-views leaves 2.8M articles.

We also remove all pages with less than 500 characters, as these pages are often low-quality stub articles, which removes a further 16% of the articles. We remove all "meta" namespace pages such as talk, disambiguation, user pages, portals, etc. as these often contain irrelevant text or casual conversations between editors.

In order to extract usable text from the wiki-markup format of the Wikipedia articles, we remove extraneous entities from the markup including table of contents, headers, footers, links/URLs, image captions, IPA double parentheticals, category tables, math equations, unit conversions, HTML escape codes, section headings, double brace templates such as info-boxes, image galleries, HTML tags, HTML comments and all other tables.

We then split the cleaned text from the pages into paragraphs, and remove all paragraphs with less than 150 characters or more than 3500 characters. Paragraphs with the number of characters between 150 to 500 were sub-sampled such that these paragraphs make up 16.5% of the final dataset, as originally done for the SQuAD dataset. Since the majority of the paragraphs in Wikipedia are rather short, of the 60M paragraphs from the final 2.4M articles, our final Wikipedia dataset contains 8.3M paragraphs.

## B    Additional Experiment on Another SQuAD Split

We also evaluate the question generation model from Zhao et al. (2018) on another data split. We call this as Test-Split3. Test Split3 is obtained by dividing the original development set in SQuAD-v1.1 into two equal halves randomly and choosing one of them as development set and the other as test set while retaining the train set in SQuAD-v1.1. As shown in Table 8, the question generation model from Zhao et al. (2018) improves the BLEU-4 score on Test-Split3 by 1.3 (w.r.t the reproduced score).

## C    Standard Errors of Evaluation in Downstream MRC Tasks

As shown in Table 9, in the case of downstream MRC results (EM/F1) which we dicussed in Section 4, for SQuAD v1.1 and SQuAD v2.0, we selected 5 model checkpoints from the same pre-training at varying numbers of pre-training steps. We then fine-tune each of these models on the final downstream data 3 times each, picked the best performing model and reported it's score. For KorQuAD, only 1 finetuning was performed with the final pre-trained model.

Table 8: Additional experiments for effectiveness of AS on SQuAD Test-Split3. We use Small-Wiki data to pre-train existing models. Those models with * are reproduced.

| Model + pre-training method | BLEU-4 score | |
| --- | --- | --- |
| | Wikipedia Test-Wiki | SQuAD v1.1 Test Split3 |
| Zhao et al. (2018) | - | 16.8 |
| Zhao et al. (2018)* | - | 16.2 |
| Zhao et al. (2018)* + AS | 6.8 | 17.5 |

Table 9: Standard errors of EM/F1 scores in downstream MRC tasks with ASGen.

| Dataset | Dev set | | Test set | |
| --- | --- | --- | --- | --- |
| | EM | F1 | EM | F1 |
| SQuAD-v1.1 | 86.1(±0.2) | 92.6(±0.1) | - | - |
| SQuAD-v2.0 | 82.3(±0.3) | 85.6(±0.2) | - | - |
| QUASAR-T(Short) | 76.6(±0.1) | 80.3(±0.2) | 76.6(±0.2) | 80.0(±0.3) |
| QUASAR-T(Long) | 74.9(±0.6) | 78.1(±0.7) | 74.0(±0.8) | 76.9(±0.9) |

