# OpenReview forum: "ASGen: Answer-containing Sentence Generation to Pre-Train Question Generator for Scale-up Data in Question Answering"
_ICLR.cc/2020/Conference — Reject_

### Official Review · AnonReviewer2 · 2019-10-20
**Official Blind Review #2**

**Rating:** 6

**Review:**

This paper proposes a pretraining technique for question generation, where an answer candidate is chosen beforehand, and the objective is to predict the answer containing sentence given a paragraph excluding this sentence and the target answer candidate. The intuition of this method is that question generation requires to generate a sentence which contains the information about the answer while being conditioned on the given paragraph. In particular, the paper compares its approach to Devlin’s presentation (https://nlp.stanford.edu/seminar/details/jdevlin.pdf according to the references; is it not a published work?) which uses next sentence generation for pretraining, that is less related to the downstream question generation task.
The proposed pretrained model is then used to finetune on a standard question generation task, which is then used to generate synthetic QA pairs for data augmentation in QA. The proposed method is evaluated on four datasets (SQuAD v1, SQuAD v2, KorQuAD, QUASAR-T) and have shown substantial performance gain.

The strength of this paper is clear to me: the idea in the paper is new and interesting, and they provide a good intuition of why the proposed learning objective is helpful compared to the previous methods.

However, I have several concerns as follows.

First, the performance improvements are marginal (less than 1.0 on three datasets, except QUASAR-T, which is actually interesting given it is under transfer learning setting).

Second, there is a very related work, “Dong et al. Unified Language Model Pre-training for Natural Language Understanding and Generation NeurIPS 2019” which also proposes a new pretraining approach for question generation and data augmentation for question answering. Not only this submission did not discuss this paper (it was posted in May 2019, which is substantially ahead of ICLR deadline), the result is substantially worse in both BLEU score of question generation and EM/F1 of end QA performance after data augmentation on SQuAD v2.

Third, the paper does not discuss any work in question generation overall, although the task of question generation was studied for decades, including question generation as data augmentation for question answering. I believe this paper should discuss previous work in question generation and compare the performance with them.

Some of the work on question generation.
- Heilman & Smith. Good Question! Statistical Ranking for Question Generation. NAACL 2010.
- Wang et al. Learning to ask questions in open-domain conversational systems with typed decoders. ACL 2018.
- Zhao et al. Paragraph-level Neural Question Generation with Maxout Pointer and Gated Self-attention Network. EMNLP 2018.
- Kim et al. Improving Neural Question Generation Using Answer Separation. AAAI 2019.

Some of the work on question generation for question answering.
- Du et al. Learning to ask: Neural question generation for reading comprehension. ACL 2017.
- Song et al. A Unified Query-based Generative Model for Question Generation and Question Answering. AAAI 2018.
- Tang et al. Learning to Collaborate for Question Answering and Asking. NAACL 2018.
- Sachan & Xing, Self-training for jointly learning to ask and answer questions. NAACL 2018.
- Tang et al. Question Answering and Question Generation as Dual Tasks. AAAI 2018.
- Lewis et al. Unsupervised Question Answering by Cloze Translation. ACL 2019.

Fourth, it looks like the paper uses the development set for both development and the final evaluation. The paper should either use a portion of the train set for development and treat the original development set as the test set, or just show the final model’s performance on the hidden test set by submitting to the leaderboard. (Especially, for Table 3, did authors reimplement BERT+NS and evaluate on the dev set? Because Devlin’s presentation only shows the result on the test set from the official leaderboard.)


(This is not a weakness of the paper, but more a subjective opinion about the paper's claim) The paper claims answer-containing sentence generation is close to question generation. However, unlike question generation where core information about the answer is given as input text, answer containing sentence generation should generate the information about the answer itself with no given information. Consider the first example in Table 5. Question generation requires to read “The racial makeup of Fresno was 245,306 (49.6%)” and generate question “What percentage of the Fresno population is White?” However, in the proposed pretraining technique, the generation model is supposed to generate “The racial makeup of Fresno was 245,306 (49.6%)” with no information. In fact, I think the fact that answer candidate is given makes it closer to question generation task, rather than “generating answer-containing sentence” is the key.

(By the way, I am giving 3 as a rating although my actual rating is closer to 4 or 5, because 4 or 5 is blocked in the review system. I am happy to increase the score after rebuttals.)

--------------------------------------------------------------------------------------------------------------------------------------------

Now, here are clarification questions.

Regarding Section 2.1
1) Did you attempt to predict ‘number of answer candidates’ by regression or classification? Which objective function did you use? Based on the current description, it looks like it’s neither regression nor classification which makes me confused.
2) I believe the second last formula in this section should say “#{...}=k” instead of “#{...}”?
3) Have you compared with thresholding instead of predicting the number of answer candidates?
4) What is the objective of training the span selection model? There are multiple candidate spans given a single sentence, and hopefully you do not want to discourage candidates except only one.
5) How was this model trained (both predicting the number and predicting the span)? Was it trained on QA datasets such as SQuAD? Then, did you train this model & go through preprocessing separately for each dataset? (Except QUASAR-I which uses synthetic data from SQuAD v1.)
6) I believe that the most popular approach in question generation literature is selecting named entities & noun phrases using off-the-shelf tools, and wonder if authors have compared their method with this baseline.

Regarding Section 2.2
1) Although this section is the most important section in the paper, the description is a bit confusing to me. My understanding is that, if the initial paragraph has three sentences, <Sentence 1>, “The racial makeup of Fresno was 245,306 (49.6%).”, and <Sentence 3>, and the answer prediction model (described in Section 2.1) chooses “49.6%”, the question generation model is given “<Sentence 1> 49.6% <Sentence 3>”, and the model is supposed to generate  “The racial makeup of Fresno was 245,306 (49.6%).”. Is it correct?
2) Did you insert any special token to indicate whether it is the sentence or the target answer candidate, or which sentence is before or after the target sentence?
3) Have you tried the formulation of the masked language model, instead of the proposed model?

Regarding Section 3
Is preprocessing the same for all Golub et al 2017, NS and AS?

Regarding Section 4
1) I am curious why the improvement in BLEU-4 score (in Table 2) is significant but the improvement in the end task (in Table 3) is marginal. Do you have any intuition on it?
2) For Figure 5(a), do you have results with BERT without any data augmentation as well? I wonder if using too small number of SQuAD annotations makes question generation inaccurate, which makes synthetic data be bad quality and hurts the performance compared to not using any synthetic data.
3) How many synthetic QA pairs are used for Table 3? Curious since the model gets around 92.5 with the largest amount of data shown in Figure 5(b) but the number reported in Table 3 is 92.8.




**** Update on Nov 10 **** Increasing the score to 6.


**Experience Assessment:**

I have read many papers in this area.

**Review Assessment: Checking Correctness Of Derivations And Theory:**

I assessed the sensibility of the derivations and theory.

**Review Assessment: Checking Correctness Of Experiments:**

I carefully checked the experiments.

**Review Assessment: Thoroughness In Paper Reading:**

I read the paper thoroughly.

---

> ### Author Response · Authors · 2019-11-09
> **Response-3/3 to AnonReviewer2**
>
> We appreciate your valuable comments and feedback.
>
> Regarding questions on Section 2.2,
>
> 1) The input is indeed <Sentence 1> 49.6% <Sentence 3> for pretraining Question Generator. However, as we mentioned in section 2.2, for fine-tuning question generator, the input is the unmodified original context.
>
> 2) We do not insert any special token to indicate the candidate answer, but we do indicate to the model the location of the target candidate answer using original BERT’s segmentation encoding with segmentation id 1 for the candidate answer tokens and segmentation id 0 for the rest of the context, as mentioned in section 2.2.
>
> 3) The encoder for the baseline model (Golub et. al.) for the question generation task is initialized with BERT which is pre-trained using masked language model. We did not try masked LM formulation for pre-training of the decoder.
>
>
>
> Regarding the question on Section 3,
>
> 1) The preprocessing is the same for all Golub et. al. 2017, NS and AS, except that for AS pre-training we use the number of candidate answers as predicted by the dynamic answer generator.
>
>
>
> Regarding questions on Section 4,
>
> 1) The question generation performance (BLEU-4) outperforms the previous work (NS) substantially on SQuAD-v1.1. The performance increase, when compared with NS on the downstream MRC task, can be smaller if the size of fine-tuning data is enough for learning, reducing the effects of pre-training. Note the large difference in performance between NS and AS in Figure 5(a) for fine-tune data of size 1.25K. However, these experiments still show consistent improvements with our method.
>
> 2) We skipped these scores in Figure 5(a) to maintain clarity of the diagram, as BERT without data augmentation performed significantly worse than both NS and AS.
> We report the F1 scores for BERT without data augmentation below -
> 1.25K: 71.1
> 5K: 81.6
> 20K: 86.5
> 87K : (Entire SQuAD) - 90.9
> We will add these scores to the manuscript.
>
> 3) As we mentioned in Section 7, we use 43M synthetic QA pairs for all experiments. However for Figure 5(b), as we mentioned in Section 4, the total number of training steps for all data sizes is kept the same as that of 10M synthetic QA pairs. The number of steps for 43M synthetic QA pairs is 4.3 times that for 10M.
>
>
> References:
> Dong et al. : Unified Language Model Pre-training for Natural Language Understanding and Generation NeurIPS 2019
> Golub, David, et al. : "Two-Stage Synthesis Networks for Transfer Learning in Machine Comprehension." Proceedings of the 2017 Conference on Empirical Methods in Natural Language Processing. 2017.
> Subramanian et. al. : "Neural Models for Key Phrase Extraction and Question Generation." Proceedings of the Workshop on Machine Reading for Question Answering. 2018.
> KorQuAD leaderboard : https://korquad.github.io/KorQuad%201.0/

---

> > ### Comment · AnonReviewer2 · 2019-11-10
> > **Thanks for your detailed responses.**
> >
> > Thanks for your detailed responses.
> >
> > - I misread the SQuAD QG evaluation comparison between UniLM and this work. I agree that this work achieves better BLEU score in QG, and QA performance are not comparable since they used different QA model. I also agree that UniLM idea and this paper's idea are orthogonal.
> >
> > - I hope my concerns about discussing Question Generation literature in the related work, and clarifying some details in Section 2 are reflected in the updated version of the paper.
> >
> > Other responses mostly resolve my concern: I am increasing my score to 6.

---

> > > ### Author Response · Authors · 2019-11-15
> > > **Update of submission based on the AnonReviewer2's feedback**
> > >
> > > Thank you for your constructive feedback and efforts.
> > >
> > > We have uploaded a new revision with the following changes based on your feedback -
> > >
> > > 1) Added the effect of our proposed pre-training with existing question generation methods Zhao et. al. 2018 and UniLM (Dong et al. 2019) achieving an increase in BLEU-4 score with both these generation methods as well. (Table 4, also Appendix-B)
> > > 2) Related question generation methods are now referenced. (Table 2, Related Works)
> > > 3) Shifted relevant tables and evaluation results to use train, dev, and test set from UniLM’s SQuAD split 1 and split 2 instead of using the development set of SQuAD.
> > > 4) Added a larger model achieving SOTA with ASGen in SQuAD question generation with BLUE-4 score of 28.0. (Table 2)
> > > 5) Added discussion on BLUE-4 score improvement for question generation v.s. the EM/F1 score improvement for downstream MRC tasks. (Section 4)
> > > 6) Clarified the objective function for training the answer generator and fixed a typo in the equation. (Section 2.1). Compared score with thresholding for predicting the number of answers. (Section 4). Referenced a previous work which explains the use of neural generation models over off-the-shelf tools for answer generation. (Related works)
> > > 7) Added results without training on synthetic data to ablation on data size. (Figure 5(a))
> > >
> > > References:
> > > Zhao et al. : paragraph-level neural question generation with maxout pointer and gated self-attention networks. EMNLP 2018
> > > Dong et al. : Unified Language Model Pre-training for Natural Language Understanding and Generation NeurIPS 2019

---

> ### Author Response · Authors · 2019-11-09
> **Response-2/3 to AnonReviewer2**
>
> We appreciate your valuable comments and feedback.
>
> Regarding questions on Section 2.1,
>
> 1) As mentioned in Section 2.1, we use regression to predict the number of answer candidates, and the objective function is MSE loss between the predicted and ground truth number of answers.
>
> 2) We will update “#{...}” to “#{...}=k” in the formula in Section 2.1. Thank you for pointing out this error.
>
> 3) We did indeed attempt thresholding the start and end logits as a first approach, but the results were worse than taking Fixed-5 top answers (baseline), with the best score of Mean Absolute Distance for thresholding was 1.01, compared to our score of 0.72, and baseline score of 0.91.
>
> 4) As mentioned in Section 2.1, we use cross-entropy loss between predicted start/end logits for each token and ground truth start/end of the span. As this loss is for each token, our method does indeed work with multiple answer candidate spans in a single input context.
>
> 5) We train all generation models using the corresponding downstream QA dataset such as SQuAD and KorQuAD as we mentioned in Section 3 Experimental Setup. We go through preprocessing separately for each dataset. (Except for QUASAR-T which uses synthetic data from SQuAD-v1.1.). Predicting the number of answers and predicting the span are jointly trained to minimize the sum of their respective losses as mentioned in Section 2.1.
>
> 6) We used Answer Generation as it was used in our baseline models of NS and Golub et. al. instead of using off-the-shelf tools in order to maintain a fair comparison, with the exception of dynamic-K. Neural models have been shown to generate better answers than using off-the-shelf tools, such as shown in Subramanian et. al. 2018. Also, our approach (Dynamic-K and AS pre-training) can be used with any form of answer generation, including those from off-the-shelf tools.

---

> ### Author Response · Authors · 2019-11-09
> **Response-1/3 to AnonReviewer2**
>
> We appreciate your valuable comments and feedback.
>
> We recently noticed the work of UniLM (Dong et al). However, the results from our paper (AS) and UniLM are comparable only in the question generation task on SQuAD-v1.1 (BLEU-4, AS vs UniLM: 25.9 vs 23.8), in which AS achieves a better BLUE score. In the case of downstream evaluation on SQuAD-v2.0, the gains from the different base MRC models by using generation methods are similar (F1, BERT+AS vs UniLM-QA+UniLM: +4.0 vs +4.2). These scores are however not comparable due to differences in the base model, the underlying data used to generate synthetic data, training methodology, etc.  Also, our AS approach can probably be used parallel to UniLM method instead of being a substitute for it.
>
> We will thoroughly read more related work papers and will update the manuscript by comparing the results from them. We cited Devlin’s presentation because as of yet this work is unpublished to the best of our knowledge.
>
> Regarding development set scores, as mentioned in Section 3, we reproduced methods of Golub et al and NS as they were described in their original work. Our BERT+AS model is already on the KorQuAD test leaderboard (named ‘BERT-CLKT-MIDDLE’), and we have also reported the results on the test set for QUASAR-T. We will also submit our model to the SQuAD leaderboard.
>
> We agree that generating sentence conditioned on a given answer is close to the question generation task. To clarify this issue, we will update the manuscript with the BLEU-4 score of generating answer-containing sentence without a given candidate answer.

---

### Official Review · AnonReviewer3 · 2019-10-27
**Official Blind Review #3**

**Rating:** 6

**Review:**

The paper in the field of machine reading comprehension. The authors address the issue of generating labeled data of question-answer tuples, without the need of manual annotation. Specifically, the authors propose a method that dynamically generates K answers given a paragraph  in order to generate diverse questions and, secondly, pre-training the question generator on answers in a sentence generation task. The authors then show that this method is superior to existing baseline methods.

Overall the paper is written rather well, however,  at times  it is tough  to understand because of the technical writing and heavy use of abbreviations.  The experiments make sense given the research question.

I have a couple of questions:

1.  Section 4&5: How often where these experiments repeated? What are the standard errors?
2. How sensitive is the architecture w.r.t. its hyper-parameters?
3. What was the training time/training cost when running the method? How does the number of parameters compare to previous state of the art?
4. In Figure 5a the F1 scores of Bert+NS and Bert+AS seem to converge with increasing  data size. Why does the difference between those two methods seem to vanish when data set size increases?


Additional comment:  "[..] researchers have proposed MRC models, often surpassing human performance."  This is a bold statement.  Machines definitely do not surpass humans in reading comprehensions.

**Experience Assessment:**

I do not know much about this area.

**Review Assessment: Checking Correctness Of Derivations And Theory:**

I assessed the sensibility of the derivations and theory.

**Review Assessment: Checking Correctness Of Experiments:**

I assessed the sensibility of the experiments.

**Review Assessment: Thoroughness In Paper Reading:**

I read the paper at least twice and used my best judgement in assessing the paper.

---

> ### Author Response · Authors · 2019-11-09
> **Response to AnonReviewer3**
>
> We appreciate your compliments and valuable feedback.
>
> 1. Regarding Section 4&5, 3 experiments were performed to compute the mean distance for K and BLEU-4 for question generation. Then the model with the maximum score was selected among them and its score is reported. Also, the following fine-tuning experiments on SQuAD were performed 3 times based on the selected model after pre-training.
>
> For table 1,
> Dynamic-K Answer Prediction: 0.73(±0.004)
>
> For table 2,
> Pre-train on Wikipedia (AS, BLEU-4) : 11.2(±0.1)
> Fine-tune on SQuAD (AS, BLEU-4) : 25.5(±0.3)
>
> In the case of downstream MRC results (F1/EM), for SQuAD v1.1 and SQuAD v2.0, we selected 5 model checkpoints from the same pre-training at varying numbers of pre-training steps. We then fine-tune each of these models on the final downstream data 3 times each, picked the best performing model and reported it’s score. For KorQuAD, only 1 finetuning was performed with the final pre-training checkpoint. We will provide the standard deviation of the performance of these models below.
>
> For table 3,
> SQuAD v1.1 (AS, F1/EM) : 92.6(±0.1) / 86.1(±0.2)
> SQuAD v2.0 (AS, F1/EM) : 85.6(±0.2) / 82.3(±0.3)
>
> For table 4,
> QUASAR-T (Long, dev-F1/EM, test-F1/EM): BERT+AS : 78.1(±0.7) / 74.9(±0.6), 76.9(±0.9) / 74.0(±0.8)
> QUASAR-T (Short, dev-F1/EM, test-F1/EM): BERT+AS : 80.3(±0.2) / 76.6(±0.1), 80.0(±0.3) / 76.6(±0.2)
>
> 2. The sensitivity of our architecture w.r.t. hyper-parameters is the same as that of the original BERT model. As we mentioned in Section 3, we used all official BERT hyper-parameters for all our models, i.e., the MRC model, the Question Generator and the Answer Generator to ensure a fair comparison with previous approaches.
>
>
> 3. The parameters/training cost of our model is as follows,
> Number of Parameters -
> Answer Generator (Golub et. al. vs NS vs AS) - 342M vs 342M vs 343M
> Question Generator (Golub et. al. vs NS vs AS) - unchanged
> MRC model (Golub et. al. vs NS vs AS) - unchanged
>
> Training cost -
> Answer Generator (Golub et. al. vs NS vs AS) - no measurable difference
> Question Generator (Golub et. al. vs NS vs AS) - unchanged compared to NS
> MRC model (Golub et. al. vs NS vs AS) - unchanged
>
> The number of parameters for the Answer Generator increases very slightly compared to previous approaches by 1M from 342M to 343M for dynamically predicting the number of answers.
>
>
> 4. Regarding Figure 5(a), with an increase in data size, more steps are needed for fine-tuning for the same number of epochs. We conjecture that this may reduce the effect of pre-training due to catastrophic forgetting, and while there are several techniques that may mitigate this effect, we did not use any of these methods to ensure a fair comparison with previous approaches. Also, if the size of fine-tuning data is enough for learning, it reduces the effects of pre-training.
>
> Moreover, except for the case of only 1.25K examples, the gap in performance between AS and NS remains roughly constant. Notice that the gap slightly increases rather than converge for an increase in data size from 20K to 87K in Figure 5(a).
>
> 5. Regarding the additional comment,
> We will clarify the statement regarding MRC model performance as follows. “[..] researchers have proposed MRC models, often surpassing human performance on these datasets.” (as is indeed true for all the datasets considered in this paper.)
>
> References:
> Golub, David, et al. : "Two-Stage Synthesis Networks for Transfer Learning in Machine Comprehension." Proceedings of the 2017 Conference on Empirical Methods in Natural Language Processing. 2017.

---

> > ### Author Response · Authors · 2019-11-15
> > **Update of submission based on the AnonReviewer3's feedback**
> >
> > Thank you for your constructive feedback and efforts.
> >
> > We have uploaded a new revision with the following changes based on your feedback -
> >
> > 1) Added discussion on BLUE-4 score improvement for question generation v.s. the EM/F1 score improvement for downstream MRC tasks. (Section 4)
> > 2) Clarified comment regarding “surpassing human performance”. (Section 1)
> > 3) Added standard deviation values for our experiments. (Appendix-C)

---

### Public Comment · ~Joachim_Bingel2 · 2019-10-31
**Interesting approach and intriguing results, but some weaknesses in presentation**

This submission presents an approach to generating question-answer pairs for data augmentation in training Question Answering/Reading Comprehension systems. This is done by generating a dynamic number of "answer" spans in a sentence, then by generating a question from each of those answers. The approach seems to yield good results both in terms of the intrinsic aptitude to generate questions that are meaningful (as compared to reference questions in standard datasets) and, more interestingly, in downstream RC settings. For the latter, the data augmentation strategy lifts performance by a considerable margin, while also outperforming another approach at data augmentation.

While the approach in itself and the experimental results are intriguing, the paper has some weaknesses with respect to the presentation. Most crucially, it makes a set of presuppositions that may result in very unclear passages, e.g. for readers that are not in a "Question Answering mindset". For instance, the paper mentions the generation of answers at a relatively early point without explaining what an "answer" in this context would be -- seeing that an answer is usually given with respect to a question, while here no question is yet available when an "answer" is generated from a sentence. The intended meaning is obvious after a while, but it does obstruct the reading and comprehension initially. Another presupposition is the reader's familiarity with BERT's [CLS] token -- if this familiarity is not given, the explanation in Section 2 may be difficult to understand. The presentation of the modelling in 2.1 and 2.2, in contrast, is reasonably clear.

The experimental protocol is generally well-motivated and meaningful, and as stated above, the results are intriguing. However, while it is encouraging to see that the experiments include a non-English dataset, it is not explicitly stated how Korean is processed here, seeing that BERT has no dedicated Korean model (apart from multilingual-BERT, but it is not stated whether this is used.) For BERT+CLKT, there is unfortunately no reference given.

Another comment on the intrinsic evaluation of the dynamic choice of K (generating the number of answers to extract from a sentence): it feels like RMSE would be a better metric here, as it would better contrast many small errors and few big errors in predicting the correct number of answers.

---

> ### Author Response · Authors · 2019-11-09
> **Response to Joachim Bingel**
>
> We appreciate your valuable compliments and feedback.
>
> As explained in section 2.1, we conjectured that when a human creates a QA problem from a given context, he/she is likely to pick an “answer-like” phrase from the context and generate a question based on that phrase. Following this hypothesis, to train answer generation module to generate an answer from the context without a question, we used data which already has labelled answer spans, such as SQuAD. Also as explained in section 2.1, The “[CLS]” token has been used primarily as a classification token in BERT model, and we used the hidden layer representation of the token for predicting K. We will clarify the above in the preceding sections.
>
> Pre-processing of Korean Wikipedia and KorQuAD is the same as that of English data except for the tokenizer for Korean. We use Mecab tokenizer and BPE (Byte Pair Encoding) for Korean to separate postposition words which do not exist in English. The structure of “BERT + CLKT” model is the same as that of original BERT except the model is pre-trained for the Korean language. Due to the absence of common pre-trained BERT for Korean, we used this model as a baseline to demonstrate the effectiveness of our method. We will include these explanations in the manuscript.
>
> We use L1 distance (Mean Absolute Error) as the metric for evaluation of predicted K as RMSE is less robust to the effect of a few large outliers.

---

### Author Response · Authors · 2019-11-15
**Update of submission based on the reviewers’ feedback**

We would like to thank all reviewers for their constructive feedback and effort in reviewing this paper.

We have uploaded a new revision and would like to summarize the changes we made to the initial submission based on the reviewers’ feedback -

The main changes are -

1) Added the effect of our proposed pre-training with existing question generation methods Zhao et. al. 2018 and UniLM (Dong et al. 2019) achieving an increase in BLEU-4 score with both these generation methods as well. (Table 4, also Appendix-B)
2) Related question generation methods are now referenced. (Table 2, Related Works)
3) Shifted relevant tables and evaluation results to use train, dev, and test set from UniLM’s SQuAD split 1 and split 2.
4) Added a larger model achieving SOTA with ASGen in SQuAD question generation with BLUE-4 score of 28.0. (Table 2)
5) Added discussion on BLUE-4 score improvement for question generation v.s. the EM/F1 score improvement for downstream MRC tasks. (Section 4)
6) Clarified comment regarding “surpassing human performance” (Section 1) and added standard deviation values for our experiments. (Appendix-C)
7) Clarified the objective function for training the answer generator and fixed a typo in the equation. (Section 2.1). Compared score with thresholding for predicting the number of answers. (Section 4). Referenced a previous work which explains the use of neural generation models over off-the-shelf tools for answer generation. (Related works)
8) Added results without training on synthetic data to ablation on data size. (Figure 5(a))

References:
Zhao et al. : paragraph-level neural question generation with maxout pointer and gated self-attention networks. EMNLP 2018
Dong et al. : Unified Language Model Pre-training for Natural Language Understanding and Generation NeurIPS 2019

---

### Decision · Program_Chairs · 2019-12-19

**Decision:**

Reject

**Comment:**

Thanks for an interesting discussion. The paper introduces a sound question generation technique for QA. Reviewers are moderately positive, with low confidence. Some issues remain unresolved, though: While the UniLM comparison is currently not apples-to-apples, for example, nothing prevents the authors from using their method to pretrain UniLM. Currently, QA results are low-ish, and it is hard to accept a paper based solely on BLEU scores (questionable metric) for question generation (the task is but a means to an end). Moreover, the authors do not really discuss how their method relates to previous work (see Review 2 and the related work cited there; there's more, e.g., [0]). I also find it a little problematic that the paper completely ignores all work prior to 2017: The NLP community started organizing workshops on question generation in 2010. [1]